# The Effect of Multiple Applications of Phosphate-Containing Primer on Shear Bond Strength between Zirconia and Resin Composite

**DOI:** 10.3390/polym14194174

**Published:** 2022-10-05

**Authors:** Awiruth Klaisiri, Apichai Maneenacarith, Nicha Jirathawornkul, Panattha Suthamprajak, Tool Sriamporn, Niyom Thamrongananskul

**Affiliations:** 1Division of Restorative Dentistry, Faculty of Dentistry, Thammasat University, Pathum Thani 12120, Thailand; 2Thammasat University Research Unit in Restorative and Esthetic Dentistry, Thammasat University, Pathum Thani 12120, Thailand; 3Faculty of Dentistry, Thammasat University, Pathum Thani 12120, Thailand; 4Department of Prosthodontics, College of Dental Medicine, Rangsit University, Pathum Thani 12000, Thailand; 5Department of Prosthodontics, Faculty of Dentistry, Chulalongkorn University, Bangkok 10330, Thailand

**Keywords:** phosphate-containing primer, resin composite, shear bond strength, zirconia

## Abstract

Occasional chipping can still occur with zirconia material despite its high strength. Emergency repairs can be accomplished using zirconia primer, adhesive agent, and resin composite when the fracture of zirconia exposes the zirconia framework. Phosphate-containing primers play an important role in zirconia surface treatment. The objective of this investigation was to evaluate the effect of multiple applications of phosphate-containing primer on shear bond strength between zirconia and resin composite. In this case, 78 zirconia discs were sandblasted by alumina particles; the zirconia was then randomized into six groups for single application and multiple applications of phosphate-containing primer according follows; group 1: no application, group 2: one application, group 3: two applications, group 4: three applications, group 5: four applications, and group 6: five applications. Adhesive was applied on the zirconia surface and the resin composite was bonded. Shear bond strength was assessed using a universal testing machine. The de-bonded surface was examined using a stereomicroscope. The shear bond strengths were statistically analyzed with one-way ANOVA and Bonferroni. Group 1 had the lowest shear bond strength with a significant difference compared to groups 2–6, whereas group 4 had the highest shear bond strength, with no significant difference compared to groups 5–6. The failure mode revealed 100% adhesive failure in all groups. In conclusion, to maximize shear bond strength at zirconia and resin composite interfaces, sandblasted zirconia surfaces should be treated with three applications of phosphate-containing primer prior to the adhesive agent.

## 1. Introduction

Zirconia is a polycrystalline solid with high strength. The three major phases are temperature-dependent: monoclinic at ambient temperature, tetragonal at roughly 1170 °C, and cubic at approximately 2370 °C, where zirconia displays phase transformation toughening characteristics, changing phase as temperature varies. At room temperature, yttria oxide (Y_2_O_3_) must be added to partially stabilize zirconia in the tetragonal phase. When stress is applied, tetragonal is transformed into monoclinic. Transformation toughening from tetragonal to monoclinic provides an increase in zirconia’s volume, preventing the fracture from propagating and improving its mechanical characteristics. Three mol% yttria was used in the first-generation zirconia. It has great strength but also high opacity due to the birefringence of the noncubic phase. However, the esthetics of second-generation zirconia were still inadequate. The third generation has a greater yttria contents, for example 4 or 5 mol%, resulting in increased non-birefringence of the cubic phase. Although the translucency has improved, zirconia undergoes less stress transformation resulting in a decreased strength and toughness [1,2].

Zirconia is slowly becoming a material of choice in today’s dentistry. Implants, orthodontic brackets, crowns and bridges, dental posts, and removable prosthodontics are made from zirconia due to its high toughness, its resistance to compression, wear, tear, and corrosion, the fact that it is lightweight, and its superior properties compared to other types of ceramic. Zirconia also provides esthetics that are similar to natural tooth color and is highly biocompatible as it does not cause allergic reactions [3].

Despite zirconia being a tough material, there have been occasional functional fractures, with chipping in about 6.25% of cases [4]. In some cases, when the fracture of zirconia is exposed to the zirconia framework [4], emergency repairs can be made with zirconia primer, adhesive agent, and resin composite [5,6]. A surface treatment protocol should be performed before the repair with resin composite. However, zirconia, a polycrystalline ceramic without a glassy phase, has weak bond strength when surface-treated such as other ceramics. An alternative surface treatment procedure must be used to enhance bond strength, since polycrystalline solids limit surface roughness from hydrofluoric acid etching under typical conditions [7]. There are three principal approaches to increase zirconia bond strength; (i) mechanical bond: sandblasting with Al_2_O_3_ particles, laser, etc., (ii) chemical bond: apply primer, silanes or universal adhesive, etc., and (iii) mechanicochemical bond: tribochemical silica coating [8].

Zirconia primer is a soluble monomer in a solvent. A monomer is composed of three components: (i) a polymerizable functional group (PG) that can react during polymerization, (ii) an adhesion promoting group (AG) that can bind to the adherend and can consist of a sulfur atom, carboxylate or phosphate group, and (iii) a connecting group, acting as a link between PG and AG, where carboxylate or phosphate-containing monomers can be bonded to dental hard tissue, and non-precious metal alloy [9]. Previous studies have shown that the application of primers that contain a phosphate monomer, including 10-Methacryloyloxydecyl dihydrogen phosphate (10-MDP), 6-methacryloxyhexylphosphonoacetate (6-MHPA) and glycerol phosphate dimethacrylate (GPDM), can improve zirconia adhesion to resin materials [10,11]. When combined with sandblasting, zirconia’s oxide layer and surface wettability was increased, resulting in a phosphate monomer being able to establish chemical bonds with the oxide on the zirconia surface [12,13].

In some circumstances, chairside zirconia defection repair is possible. The most common procedure for achieving high zirconia bond strength was the use of sandblasting to obtain a mechanical bond, followed by the use of a zirconia primer and bonding agent [13]. Mahgoli et al. concluded that the bonding ability of the zirconia and resin composite was improved by using a single application of zirconia primer [14]. Moreover, Klaisiri et al. reported that when zirconia was repaired by using sandblasting and an application of 10-MDP universal adhesive, it showed increased bonding ability between zirconia and resin composite. [5]. Many studies applied only a single application of primer during zirconia bonding, which was shown to increase zirconia bond strength [14,15,16,17]. Additional multiple applications of zirconia primer may have a synergistic effect on zirconia’s chemical bonding due to an increased amount of functional monomer after application. However, there have not been any studies that have investigated whether multiple applications of primer would affect the bonding ability of zirconia and resin composite. As a result, the aim of this investigation was to evaluate the effect of multiple applications of phosphate-containing primer on the shear bond strength of zirconia and resin composite. The investigation’s null hypothesis was that the multiple application of phosphate-containing primer has no influence on zirconia and resin composite shear bond strength.

## 2. Materials and Methods

### 2.1. Specimen Preparation

A pre-sintered zirconia disc (Ceramill Zolid HT + Preshades, Amann Girrbach, Austria) was sectioned into 78 rectangular specimens, each measuring 6 × 4 × 3mm^3^. The manufacturer’s instructions were followed for sintering the zirconia specimens. Zirconia samples were positioned 1 mm above a PVC tube that contained type IV gypsum. The sample was polished using silicon carbide paper with a 600-grit (3M Wetordry abrasive sheet, USA) on an automated polishing device (Tegramin-25, Struers. Inc., Cleveland, OH, USA) at 2 kg/cm^2^ force, 100 rounds/min, under running water for 120 s.

### 2.2. Sandblasting

A zirconia specimen was subjected to sandblasting using aluminum oxide (Al_2_O_3_) with 50 μm particles for 10 s at 0.2 megapascals (MPa) pressure perpendicular to the zirconia surface (A10723 Base 3, Dental Vision Co. Ltd., Bangkok, Thailand). Running water was used to rinse off sandblasted debris on the specimens. A 20-min ultrasonic cleaning procedure was performed on the samples in distilled water. (WUC-D22H, DKSH Singapore Pte Ltd., Singapore).

### 2.3. Primer Application

A phosphate-containing primer (Clearfil ceramic primer plus (CPP), Kuraray Noritake, Tokyo, Japan) was used (Table 1), and then zirconia specimens were randomized into 6 groups of 13 specimens each, as follows:

Group 1: no primer (control group)

Group 2: one application of primer

Group 3: two applications of primer

Group 4: three applications of primer

Group 5: four applications of primer

Group 6: five applications of primer

One-sided tape (Scotch blue painter’s tape, 3M, Maplewood, MN, USA) with a thickness of roughly 80 μm was cut to a width and length of 10 mm, and a 2 mm-diameter hole was then punched. To make it simple to remove, a circular hole was reached by cutting the side of the tape. After one-sided tape was attached to the zirconia surface, primer was applied onto the zirconia surface with disposable micro brush (Applicator tips, Dentsply DeTrey, Konstanz, Germany), then the excess primer built up around the border of the tape loop was blotted off using a new micro brush. After 1 min, air was blown from a triple syringe, that was free from water and oil mist, from a mobile dental unit (Mobile dental unit, Thai Dental Products, Bangkok, Thailand) with a pressure of 40–50 pounds per square inch at a distance of approximately 10 mm from the zirconia surface, until an absence of movement of the primer was observed and it was totally dry. The process was repeated until the number of primers in each group had been applied.

### 2.4. Adhesive Agent Application

The zirconia surface was coated with adhesive agent (OptiBond S, Kerr Corporation, Orange, CA, USA) using a disposable micro brush (Table 1). The extra adhesive that had accumulated around the edge of the tape loop was then removed using a new micro brush. Air was blown from a triple syringe, which was free from water and oil mist, from a mobile dental unit with a pressure of 40–50 pounds per square inch at a distance of approximately 10 mm from the zirconia surface until an absence of movement of the adhesive was observed. Light curing (Elipar Freelight 2 LED, 3M ESPE, USA) was carried out for 20 s perpendicular to the zirconia surface.

### 2.5. Resin Composite Application

A template (Ultradent product, Inc., South Jordan, UT, USA) with a height of 2 mm and a diameter of 2 mm was fitted to the zirconia surface by aligning the hole of the template to the hole of the one-sided tape; the resin composite (SimpliShade universal nanohybrid universal restorative composite, Kerr Corporation, USA) was then pushed into the template until full (Table 1). The template and one-sided tape were removed after the resin composite had been light-cured for 40 s in a perpendicular position and as near the template as possible. The specimens were light cured for another 40 s perpendicular to the resin composite on each side, rotating a full 360 degrees. The sample was submerged in distilled water for 24 h at 37 °C.

A universal testing device (AGS-X 500N, Shimadzu Corporation, Kyoto, Japan) was used to evaluate the shear bond strength of the specimens. The shearing blade was positioned parallel to the intersection of the zirconia and resin composite. A shear pressure of 0.5 mm per min was set until fracture occurred (Figure 1). The shear bond strength in MPa was calculated by the maximum shear bond strength divided by the surface region of the bonding interface.

### 2.6. Fracture Mode Analysis

The fractured surface of the specimen was examined using a stereomicroscope at a magnification of 15× (ML9300, Meiji Techno Co. Ltd., Saitama, Japan) to determine the mode of failure, which is divided into 3 types. 

(1) Adhesive failure happens, when zirconia and resin composite fail to adhere to one another. This occurs when the zirconia surface has no resin composite remnants. 

(2) Cohesive failure happens, when the failure occurs within the resin composite. This occurs when the entire surface of the zirconia is coated with resin composite.

(3) Mixed failure happens, when adhesive and cohesive failures occur together. This happens when a remnant of resin composite remains adhered to the zirconia surface.

### 2.7. Statistical Analysis

IBM SPSS Statistics 25.0 (SPSS Inc., Chicago, IL, USA) was used to determine the quantitative data from 6 independent groups with a confidence level of 95%. The Kolmogorov-Smirnov test was used to determine normality, whereas the normal distribution was tested with one-way ANOVA and Post-hoc with Bonferroni.

## 3. Results

In this study, no specimens were de-bonded before shear bond strength was tested (prematurely failed specimen) in all the groups.

In Table 2, the shear bond strengths’ means and standard deviations are displayed. It was discovered that group 1 had the lowest shear bond strength, with a significant difference compared to groups 2–6 (*p* < 0.05). Group 4 had the highest shear bond strength, with no significant difference compared to group 5 and group 6 (*p* > 0.05).

The failure mode was as indicated in Table 2. All specimens displayed adhesive failure in all groups (Figure 2). No instances of cohesive or mixed failure were found.

## 4. Discussion

The aim of this investigation was to evaluate the effect of multiple applications of phosphate-containing primer on the shear bond strength of zirconia and resin composite. It was observed that three applications established the highest shear bond strength, which was significantly different from one application and two applications. However, applying more than three applications showed no statistically significantly difference in shear bond strength compared to three applications. Therefore, the null hypothesis was rejected.

Currently, zirconia surface treatment before repairing restoration can be performed using several methods. The most common method of gaining the best retention is to obtain both mechanical and chemical retention. Sandblasting is the most common surface treatment for obtaining mechanical retention of zirconia [13]. Chemical retention can be gained with the application of phosphate functional monomers on a sandblasted zirconia surface. At present, phosphate functional monomers can be found in either primers or adhesive agents, or in both. Shafiei et al. found that using an adhesive combined with a 10- MDP-containing primer leads to a statistically significant increase in shear bond strength. The interface was stable since the adhesive was applied and the light cure occurred, resulting in increased shear bonding ability between the zirconia and the resin composite [18]. According to Klaisiri et al., the combination of alloy primer and adhesive produced a greater shear bond strength than either alloy primer or adhesive alone [19]. This was consistent with the past study, in which the shear bond strength was greater in the primer combined with adhesive group than in the adhesive-only group.

The main zirconia bonding monomer in use has been 10-MDP as it promotes bonding with both resin composite and zirconia [20]. This is a functional monomer that contains a phosphate group at one end, which chemically bonds with zirconia, and a polymerizable methacrylate group at the other end, that bonds with the resin material. In this study, CPP was used. CPP is a phosphate-containing primer composed of 10-MDP and a silane coupling agent. According to Chuang et al., the use of silane coupling agent alone did not enhance the shear bond strength or bond durability of zirconia [16]. Furthermore, a combination of 10-MDP and silane coupling agent, did not show any synergistic effect on zirconia [21,22]. However, several studies have demonstrated that silane coupling agents are ineffective for chemically bonding non-silica-based ceramics such as zirconia [23,24].

Based on the results, it was found that one application and two applications of phosphate-containing primer prior to the adhesive agent performed better than adhesive alone. This could be because 10-MDP, a phosphate monomer, was chemically bonded to the outermost surface of zirconia [25]. A phosphate group at one end of 10-MDP can be chemically bonded to the zirconia’s oxide layer and a double-bonded methacrylate, at the other end, can polymerize with resin composite/cement, resulting in increased bond strength [26]. The shear bond strength was highest when three applications of phosphate-containing primer were combined with an adhesive. This might be due to the increase in functional monomer concentration when multiple applications of primer were applied. Thus, more bonds between phosphate and zirconia oxide are formed, improving the bonding between zirconia and resin composite [25,26]. Another reason might be the higher amount of solvent evaporation when more applications of primer were applied. The CPP used in this study contains 80% ethanol as a solvent, which evaporated as air was blown [25]. The increased multiple applications may improve the evaporation of solvent from the zirconia surface as more air was blown. This allowed the zirconia bond interface to be solvent-free providing the optimal environment for bonding; thus, improving the bonding ability between zirconia and resin composite. However, four and five applications of phosphate-containing primer with adhesive contributed to no statistically significantly different further increase in shear bond strength compared to three applications. This might be due to the complete evaporation of solvent when air was blown after three applications. Another possible reason might be that the phosphate functional monomers have bonded with all of the zirconia oxide layer present in the bonded area. This could result in no significant difference in shear bond strength when more than three applications of phosphate-containing primer were applied.

The de-bonded surface was examined under a stereomicroscope after shear bond strength inspection. All groups were determined to have had adhesive failure as their mode of failure. Therefore, the shear bond strength achieved in this investigation, with all adhesive failures, was the true strength of the zirconia and resin composite interface and was considered to be a true indication of bond strength between these materials [27,28].

This study has limitations. The incubated specimen could only interpret the shear bond strength of zirconia and resin composite 24 h after bonding. Thermocycling could be performed to evaluate the durability of bonding between zirconia and resin composite to assess long-term adhesion in a future investigation.

## 5. Conclusions

To maximize shear bond strength at zirconia and resin composite interfaces, sandblasted zirconia surfaces should be treated with three applications of phosphate-containing primer prior to the adhesive agent.

## Figures and Tables

**Figure 1 polymers-14-04174-f001:**
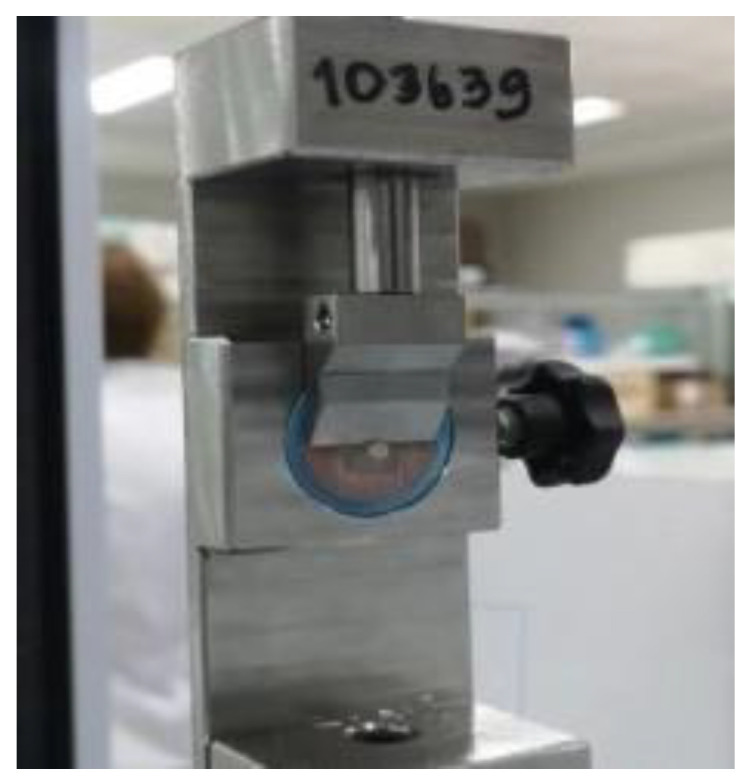
Test setup for shear bond strength.

**Figure 2 polymers-14-04174-f002:**
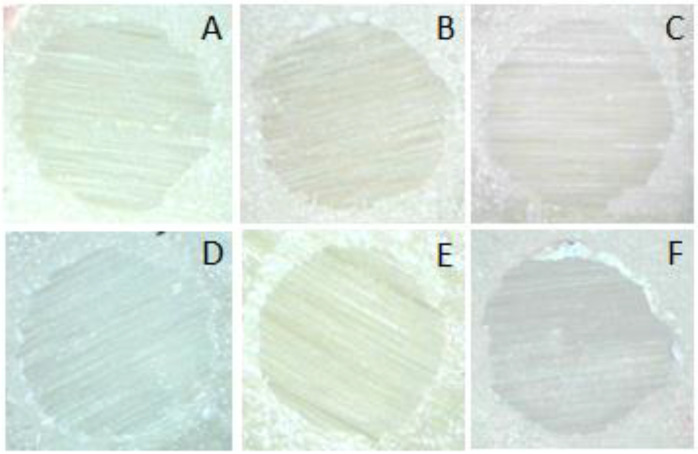
Adhesive failure mode illustrating; (**A**) group 1; (**B**) group 2; (**C**) group 3; (**D**) group 4; (**E**) group 5; and (**F**) group 6.

**Table 1 polymers-14-04174-t001:** The materials used in this study.

Material	Composition
Zirconia (Ceramill Zolid HT + Preshades, Amann Girrbach, Austria) Lot: 1905000	ZrO_2_ + HfO_2_ + Y_2_O_3_: ≥99 Y_2_O_3_: 6.0–7.0, HfO_2_: ≤5, Al_2_O_3_: ≤0.5 Other oxides: ≤1
Clearfil ceramic primer plus (Kuraray Noritake Dental, Japan) Lot: 310073	10-Methacryloyloxydecyl dihydrogen phosphate, ethanol, 3-trimethoxysilylpropyl methacrylate
Optibond S (Kerr Corporation, Orange City, CA, USA) Lot: 8811346	Ethanol, glyceryl dimethacrylate, 2-yydroxyethyl methacrylate, pyrogenic (fumed) amorphous silica, alkali fluorosilicates (Na)
SimpliShade universal nanohybrid universal restorative composite medium shade (Kerr Corporation, USA) Lot: 8129352	2,2′-ethylenedioxydiethyl dimethacrylate, poly(oxy-1,2-ethanediyl), α,α’-[(1-methylethylidene)di-4,1-phenylene]bis[ω-[(2-methyl-1-oxo-2-propen-1-yl)oxy]-, ytterbium fluoride

**Table 2 polymers-14-04174-t002:** Mean shear bond strength, standard deviation (MPa) and percentage of failure mode.

Groups	Mean Shear Bond Strength (SD)	Percentage of Failure Mode		
		Adhesive	Cohesive	Mixed
Group 1(No primer)	12.12 (3.37) ^a^	100	0	0
Group 2 (1 application)	21.77 (3.88) ^b^	100	0	0
Group 3 (2 applications)	20.37 (3.50) ^b^	100	0	0
Group 4 (3 applications)	28.48 (3.75) ^c^	100	0	0
Group 5 (4 applications)	26.34 (3.49) ^c^	100	0	0
Group 6 (5 applications)	26.17 (3.38) ^c^	100	0	0

The same letter indicates that there was no statistically significant difference.

## Data Availability

Not applicable.

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
