# Peer review of "The Effect of Multiple Applications of Phosphate-Containing Primer on Shear Bond Strength between Zirconia and Resin Composite"

_polymers, 2022, doi:10.3390/polym14194174_

Round 1

Reviewer 1 Report

The study presented in the manuscript aim to test the opportunity of using more than one layer of primer when conditioning zirconia for layering with composite resin. The study protocol is well described, the methodology is detailed and the results are clearly presented. 

The shortcoming is the narrow range of problems presented; however, the clinical importance of the problems exposed, as well as the extensive explanations based on the composition of the materials increase the value of the paper. Thermocycling would increase also the clinical relevance of the results- but it’s absence was mentioned by the authors. 

Introduction- I suggest to clarify that the chipping phenomenon refers mainly to the layered zirconia, at the interface between the glassy ceramics and zirconia. These are the situations when reparative interventions with composite are needed. No reference to the respective clinical situation has been made.

Reviewer 2 Report

Dear Authors

In attach you will find some minor remarks

Best regards
